Association between gut microbiota and allergic rhinitis: a systematic review and meta-analysis

Li Mengyao 1
Wang Qian 1 2
Wang Ruikun 3
Pu Jian 1
Zhang Yimin 1
Ye Siyu 1
Liang Jieqiong 1
Li Tao socott@126.com 4
Gu Qinglong gql13146836613@163.com 1
1 Department of Otolaryngology-Head and Neck Surgery, Capital Institute of Pediatrics , Beijing , China
2 Graduate School of Peking Union Medical College , Beijing , China
3 Capital Institute of Pediatrics-Peking University Teaching Hospital , Beijing , China
4 Child Health Big Data Research Center, Capital Institute of Pediatrics , Beijing , China
Neumann Bernd
Electronic publication date: 2025 May 26
Publication date: 2025
Volume: 13
Electronic Location ID: e19441
Received 2024 Dec 9; Accepted 2025 Apr 16
Copyright: ©2025 Li et al.
Copyright year: 2025
Copyright holder: Li et al.
License: This is an open access article distributed under the terms of the Creative Commons Attribution License, which permits unrestricted use, distribution, reproduction and adaptation in any medium and for any purpose provided that it is properly attributed. For attribution, the original author(s), title, publication source (PeerJ) and either DOI or URL of the article must be cited.
License URL: https://creativecommons.org/licenses/by/4.0/

Keywords: Gut microbiota, Allergic rhinitis, Meta-analysis, Systematic review

Funding: Key Program of Capital’s Funds for Health Improvement and Research 2022-1-2101 Beijing Municipal Natural Science Foundation 7232010 This work was supported by the Key Program of Capital’s Funds for Health Improvement and Research (2022-1-2101) and the Beijing Municipal Natural Science Foundation (7232010). The funders had no role in study design, data collection and analysis, decision to publish, or preparation of the manuscript.

==============================
Background

Many studies have shown that allergic rhinitis (AR) is closely related to intestinal flora, and probiotics are effective in treatment. However, the results of human observational studies on the correlation between intestinal flora and AR have been contradictory. The aim of this study was to determine the relationship between gut microbiota and allergic rhinitis and to provide a clinical reference.

Methods

PubMed, Web of Science, Medline, Embase, Cochrane Library, and Cinahl databases were searched, and the literature on the correlation between allergic rhinitis and the gut microbiota reported from database establishment to December 2023 was included. Literature meeting the inclusion criteria was screened, and meta-analysis of the included literature was performed using R software (4.3.3). Literature quality underwent assessment utilizing the Newcastle-Ottawa Quality Assessment Scale. Hedge’s g standardized mean difference (SMD), confidence intervals (CIs), and heterogeneity (I2) for alpha diversity were calculated. Median interquartile range (IQR) were calculated as effect statistics for the abundance of bacteria. The Egger test determined publication bias in the literature.

Results

A total of 10 observational studies in humans were conducted, identifying 550 patients with AR and 385 healthy individuals. No statistically significant differences were observed in alpha diversity between two groups, including Shannon index (SMD = −0.3938, 95% CI [−0.9847–0.1972], I2 = 94%), Simpson index (SMD = −0.16, 95% CI [−1.12–0.80], I2 = 96%) and Chao1 index (SMD = −0.00, 95% CI [−1.32–1.32], I2 = 97%). We performed a meta-analysis for the following four phyla, but found no significant differences: Firmicutes (95% CI [−0.10–0.19], I2 = 75%), Bacteroidetes (95% CI [−0.42–0.19], I2 = 95%), Proteobacteria (95% CI [−0.06–0.03], I2 = 92%), Actinobacteria (95% CI [−0.09–0.03], I2 = 83%).

Conclusions

The currently available evidence does not suggest that patients with allergic rhinitis may have similar intestinal flora imbalances. Nevertheless, further corroboration is required with larger samples and higher-quality studies.

Introduction

Allergic rhinitis (AR) represents one of the most prevalent public health problems worldwide, placing an increasing burden on individuals and society at large. The symptoms of AR include nasal congestion, watery eyes, a runny nose, and symptoms such as itching and sneezing, which have a significant impact on the quality of life of patients (Skoner, 2001). Furthermore, AR can have a broader impact on human health, leading to complications such as sleep disturbances, reduced productivity and overall quality of life (Zhang & Akdis, 2022). Studies conducted in different countries have demonstrated that the prevalence of AR ranges from 11.8% to 46.0% (Licari et al., 2023; Bousquet et al., 2020a). This results in significant healthcare costs and lost productivity.

The human gut microbiome comprises over 1,500 distinct microbial species, comprising different bacteria classified by phyla, order, class, family and genus. Firmicutes and Bacteroidetes are the dominant phyla, accounting for 90% of the total community, followed by Actinobacteria and Proteobacteria (Blaser, 2014). Bacterial communities are subject to dynamic changes and dysbiosis, and ecological imbalances in the gut microbiota can result in alterations to abundance, diversity, and function. The gut is also a pivotal organ in the pathogenesis of various extraintestinal organ diseases. For instance, trimethylamine-N-oxide (TMAO) has been identified as the primary diet-induced metabolite produced by the gut microbiota. It has been associated with an elevated risk of atherosclerotic cardiovascular disease (ASCVD) and related complications, including cardiovascular mortality or major adverse cardiovascular events (MACE) (Guasti et al., 2021).

The pathogenesis of AR is complex, with both genetic and environmental factors involved (Bousquet et al., 2020b). The relationship between the gut microbiota and AR has recently emerged as a prominent area of research. Recent years have seen an increase in research focusing on the lung-gut axis. A number of scholars have proposed that the lung-gut epithelium has the same embryological origin. When the intestinal flora and its metabolites are altered, the local intestinal immune system produces a variety of inflammatory mediators and immune cells through the circulatory system to regulate the immunity of the airway epithelium, including the lung (Shu et al., 2007). Indeed, some studies have now demonstrated that an imbalance of gut flora is strongly associated with allergic rhinitis. In animal models, the intestinal microbiota characteristics of allergic rhinitis mice were altered in comparison with healthy mice. This resulted in a significant increase in the proportion of Firmicutes and Proteobacteria, while the proportion of Bacteroidetes was significantly decreased (Chen et al., 2023). Furthermore, fecal microbiota transplantation therapy has been shown to reverse intestinal dysbiosis and protect AR mouse models (Dong et al., 2024). The results of existing observational control studies have found that allergic rhinitis (AR) is associated with alterations in the gut microbiota. Consequently, the objective of our meta-analysis was to investigate the alterations in the gut microbiota by examining the variations in the gut microbiota composition between AR patients and healthy individuals.

Methods

This study was preregistered with PROSPERO (CRD42023488136) and conducted and reported according to Preferred Reporting Items for Systematic Reviews and Meta-Analyses (PRISMA) guideline (Hutton et al., 2015). Portions of this text were previously published as part of a preprint (https://www.researchsquare.com/article/rs-3926493/v1) (Wang et al., 2024).

Search strategy

A search string was developed to identify studies reporting gut microbiota and AR (Table S1). In brief, we performed an extensive search of PubMed, Web of Science, Medline, Embase, Cochrane Library, and Cinahl databases for articles published before December 26, 2023 (last update), that contained “allergic rhinitis”and “gut microbiota” in the title subheadings and main text. To ensure comprehensive coverage, we also conducted a manual search of reviews and published references. Each database was last searched on December 26, 2023. EndNote software version X9 was used to manage the retrieved literature.

Eligibility criteria

Eligibility criteria for original studies included the following: (1) utilization of an observational case-control design, (2) implementation of gut microbiota analysis with reporting of diversity or abundance metrics, and (3) longitudinal studies featuring baseline comparisons between allergic rhinitis (AR) patients and healthy controls. Studies were excluded if they involved interventional designs or longitudinal comparisons without the inclusion of a control group.

Exclusion criteria

The exclusion criteria were as follows: (1) studies with inadequate data, (2) studies lacking a control group, (3) non-research publications such as reviews, guidelines, letters, and commentaries, and (4) redundant articles with incomplete data. Studies failing to meet these requirements were excluded.

Literature screening

Records were independently screened by two authors (MYL and RKW). Titles and abstracts that met the inclusion criteria were chosen for a full-text examination. In cases of discrepancies regarding inclusion, the two researchers discussed and reached a consensus based on their expertise and access to information, or sought third-party experts’ opinion for evaluation, and finally, Professor QL Gu made the final recommendation.

Data extraction

Data from each eligible article were initially extracted by one researcher using a customized data table. Afterward, another researcher conducted a review to ensure the accuracy of the extracted data. In case of discrepancies during data extraction, consensus was reached through discussion or, if necessary, consultation with a third-party expert. For data extraction, Microsoft Excel from Microsoft Corp. in Redmond, WA, United States, was employed. The following aspects were included:

1. Basic information: this contained article’s title, author, publication year, and study location.

2. Study characteristics: this included total number of participants meeting inclusion criteria, their age, gender and BMI.

3. Experimental methods: this included diagnostic criteria for allergic rhinitis and microbiological evaluation techniques.

4. Study outcomes: this involved the index of Shannon, Simpson and Chao1; gut microbiota (GM)’s relative abundance of phylum levels. Contacted the corresponding author or used Engauge Digitizer for digital processing to obtain sufficient data when necessary.

Quality assessment

Literature quality underwent assessment by two researchers utilizing the Newcastle Ottawa Quality Assessment Scale (NOS), with a third reviewer aiding in resolving discrepancies when necessary (Lo, Mertz & Loeb, 2014). The NOS encompasses three criteria: selectivity, comparability, and exposure. Each study is granted a maximum of nine stars. Studies achieving scores greater than or equal to 7 were categorized as possessing good quality, while those scoring between 5 and 6 were considered average, and those scoring 0 to 4 were deemed poor in quality.

Statistical analysis

All the analysis were performed using the R software (4.3.3). Two researchers independently analyzed the data. The meta-analysis was conducted using the “meta” and “meta median” packages. The alpha diversity between AR patients and controls was analyzed in studies where sufficient data were reported. For alpha diversity data that could be expressed as mean and standard deviation, standardized mean differences (SMD) with 95% confidence intervals (95% CI) were employed as effect measures. A random-effects meta-analysis was performed on the SMD using the inverse-variance method. For the abundance of bacteria that are not evenly distributed, we used median interquartile range (IQR) as effect statistics. Inter-study heterogeneity was quantified using the DerSimonian-Laird estimator, reported with the I2 statistic. Significant heterogeneity was defined as I2 ≥ 50% and P < 0.05. Based on a systematic review of interventions by Cochran’s Handbook, the I2 value can be interpreted as follows: when the I2 value is between 0% and 40%, it may indicate that heterogeneity is not important. In cases where the I2 value is between 40% and 75%, it indicates a moderate degree of heterogeneity. I2 values in the range of 75% to 100% indicate a considerable degree of heterogeneity. Pre-planned subgroup and meta-regression analyses were performed to explore and determine the potential sources of heterogeneity. The assessment of publication bias was conducted using the Egger linear regression test. Sensitivity analysis was conducted by removing the high-risk studies.

Results

Study selection

According to the search flow chart, a total of 6,205 articles were retrieved. Among them, 1,003 were duplicates and 5,202 required preliminary screening. Following a comprehensive review of the titles and abstracts, a total of 5,151 articles were excluded on the grounds of animal experiments, reviews, patents and other criteria. Finally, 10 eligible studies were included, six from PubMed database and four from Web of science database (Chiu et al., 2019a; Zhu et al., 2020; Liu et al., 2020; Watts et al., 2021; Zhou et al., 2021; Sahoyama et al., 2022; Lin, Li & Li, 2022; Chiu et al., 2023; Zhang et al., 2023; Wan et al., 2023). Figure 1 shows the detailed selection process. Of these included studies, 10 provided indicators related to α diversity. This meta-analysis included seven studies comparing the relative abundance of taxa.

Figure 1 PRISMA flow diagram.

Study characteristics, risk of bias of included studies

Table 1 presents a summary of the study characteristics. The included studies were published between 2019 and 2023, with a total of 10 studies (Chiu et al., 2019a; Zhu et al., 2020; Liu et al., 2020; Watts et al., 2021; Zhou et al., 2021; Sahoyama et al., 2022; Lin, Li & Li, 2022; Chiu et al., 2023; Zhang et al., 2023; Wan et al., 2023) and 935 individuals. These studies covered a total of three countries. The sample size of AR patients in all studies ranged from 18 to 186, and most studies (7, 70%) had a small sample size (less than 60) (Chiu et al., 2019a; Zhu et al., 2020; Watts et al., 2021; Zhou et al., 2021; Chiu et al., 2023; Zhang et al., 2023; Wan et al., 2023). In assessing the microbiome, 16S rRNA gene sequencing technology was employed in seven studies (Chiu et al., 2019a; Zhu et al., 2020; Liu et al., 2020; Watts et al., 2021; Zhou et al., 2021; Sahoyama et al., 2022; Zhang et al., 2023), while metagenomic shotgun sequencing technology was utilized in three studies (Lin, Li & Li, 2022; Chiu et al., 2023; Wan et al., 2023). Noticeable disparities were observed among studies in terms of alpha diversity indices. Significant distinctions were noted between fecal disposal and DNA extraction methodologies (Table S2).

Table 1 Key features of the literature encompassed in the systematic review.

Study/Location	Sex (Female/male)	Age (Year, M ± SD)	BMI (Kg/m2, M ± SD)	Measure	Measurement method	
	AR	HC	AR	HC	AR	HC			
Chiu et al. (2019a), Chiu et al. (2019b)/China	11/17	12/14	5.9 ± 0.9	5.6 ± 0.8	NA	NA	Shannon, Chao1	16S rRNA gene sequencing	
Zhu et al. (2020)/China	22/11	13/18	31.79 ± 9.91	32.06 ± 9.26	21.86 ± 3.16	21.85 ± 2.57	Shannon, Simpson, Chao1, GM abundance	16S rRNA gene sequencing	
Liu et al. (2020)/China	40/53	30/42	29 ± 15.59	24.47 ± 19.46	NA	NA	Shannon, Simpson, Chao1, GM abundance	16S rRNA gene sequencing	
Watts et al. (2021)/Australia	35/22	13/10	39.06 ± 13.29	36.55 ± 10.51	25.96 ± 4.08	24.25 ± 4.61	Shannon, Chao1	16S rRNA gene sequencing	
Zhou et al. (2021)/China	12/6	8/9	30.6 ± 1.6	33.1 ± 2.3	22.6 ± 0.8	22.3 ± 0.4	Shannon	16S rRNA gene sequencing	
Sahoyama et al. (2022)/Japan	23/163	9/97	49.2 ± 7.1	50.4 ± 8.2	23.6 ± 3.0	23.7 ± 3.1	Shannon, Chao1	16S rRNA gene sequencing	
Lin, Li & Li (2022)/China	19/49	17/21	6.67 ± 2.45	6.79 ± 2. 40	16.9 ± 3.9	16.8 ± 2.5	Shannon, Simpson, Chao1	shotgun metagenomic sequencing	
Chiu et al. (2023)/China	11/14	14/14	5.8 ± 0.9	5.7 ± 0.8	NA	NA	Shannon	shotgun metagenomic sequencing	
Zhang et al. (2023)/China	9/15	10/15	7.42 ± 2.89	6.72 ± 2.56	15.98 ± 3.58	17.36 ± 4.27	Shannon, Simpson, Chao1, GM abundance	16S rRNA gene sequencing	
Wan et al. (2023)/China	8/10	8/11	8.9	9.5	17.8	18.6	Shannon, Simpson, GM abundance	shotgun metagenomic sequencing	

Quality assessment

Detailed results of the methodological quality of the study are shown in Table 2. Among the corpus of 10 studies scrutinized, all received a commendation for their high quality.

Table 2 Research quality rating scale.

Study	Selection	Comparability	Exposure	Score	
	A	B	C	D	E	F	G	H	I		
Chiu et al. (2019a), Chiu et al. (2019b)	✓	✓		✓	✓	✓	✓	✓	✓	8	
Zhu et al. (2020)	✓	✓		✓	✓		✓	✓	✓	7	
Liu et al. (2020)	✓	✓		✓	✓		✓	✓	✓	7	
Watts et al. (2021)	✓			✓	✓		✓	✓	✓	6	
Zhou et al. (2021)	✓	✓		✓	✓		✓	✓	✓	7	
Sahoyama et al. (2022)		✓	✓		✓		✓	✓	✓	6	
Lin, Li & Li (2022)	✓	✓	✓	✓	✓		✓	✓	✓	8	
Chiu et al. (2023)				✓	✓		✓	✓	✓	5	
Zhang et al. (2023)	✓	✓	✓	✓	✓		✓	✓	✓	8	
Wan et al. (2023)	✓	✓	✓	✓	✓		✓	✓	✓	8	
Notes.

A, adequate case definition or representativeness of the exposed cohort; B, representativeness of cases or selection of non-exposed queues; C, selection of controls or determination of exposure; D, definition of controls or no outcome event occurred before the start of the research object; E, controlled for age; F, controlled for additional factors; G, ascertainment of exposure or evaluation of outcome events; H, same method for cases and controls or adequacy of follow-up; I, non-response rate or integrity of follow-up.

The results of the meta-analysis

Alpha diversity

A total of ten studies (Chiu et al., 2019a; Zhu et al., 2020; Liu et al., 2020; Watts et al., 2021; Zhou et al., 2021; Sahoyama et al., 2022; Lin, Li & Li, 2022; Chiu et al., 2023; Zhang et al., 2023; Wan et al., 2023) provided data on the Shannon index, five studies (Zhu et al., 2020; Liu et al., 2020; Lin, Li & Li, 2022; Zhang et al., 2023; Wan et al., 2023) provided data on the Simpson index and seven (Chiu et al., 2019a; Zhu et al., 2020; Liu et al., 2020; Watts et al., 2021; Sahoyama et al., 2022; Lin, Li & Li, 2022; Zhang et al., 2023) studies provided data on the Chao1 index. Four of the studies provided all three of the requisite elements. Zhang’s study yielded results that were contrary to the majority of the other studies with respect to the Shannon and Simpson indices (Zhang et al., 2023). The direction of differences was consistent across indices within other studies. However, the direction of differences was not consistent across studies. The combined effect demonstrated no discernible difference between the AR patient and control groups. The main results of alpha diversity are shown in Table S3.

The Shannon index was reported on in 10 studies (Chiu et al., 2019a; Zhu et al., 2020; Liu et al., 2020; Watts et al., 2021; Zhou et al., 2021; Sahoyama et al., 2022; Lin, Li & Li, 2022; Chiu et al., 2023; Zhang et al., 2023; Wan et al., 2023), which presented findings on changes in this index in the intestinal flora of patients with AR. Of the studies in question, two reported a significantly higher Shannon index in AR patients compared to healthy individuals, while four indicated a lower index and four reported no significant differences. Given the considerable heterogeneity among the studies (I2 = 94%, P < 0.05), we adopted the random effects model for analysis. The results demonstrated no statistically significant difference in the Shannon index between patients with allergic rhinitis and healthy individuals (SMD = −0.3938, 95% CI [−0.9847–0.1972]) (Fig. 2A). The results of the Egger test indicated a low probability of publication bias (t = −0.18, p = 8648).

Figure 2 Forest plot of the Alpha diversity.

(A) Shannon. (B) Simpson. (C) Chao1.

The Simpson index was reported on by five studies (Zhu et al., 2020; Liu et al., 2020; Lin, Li & Li, 2022; Zhang et al., 2023; Wan et al., 2023), which presented findings regarding changes in this index in the intestinal flora of patients with allergic rhinitis. Of the studies included in this review, one reported a significantly higher Simpson index in AR patients compared to healthy individuals, while another found it to be lower. The remaining three studies did not identify any significant differences. Given the considerable heterogeneity among the studies (I2 = 96%, P < 0.05), we employed the random effects model for the analysis. The results demonstrated no statistically significant difference in the Simpson index between patients with allergic rhinitis and healthy individuals (SMD = −0.16, 95% CI [−1.12–0.80]) (Fig. 2B).

The Chao1 index was reported to have undergone changes in the intestinal flora of AR patients by seven studies (Chiu et al., 2019a; Zhu et al., 2020; Liu et al., 2020; Watts et al., 2021; Sahoyama et al., 2022; Lin, Li & Li, 2022; Zhang et al., 2023). Of the studies in question, one reported a significantly higher Chao1 index in AR patients compared to healthy individuals, while two studies indicated a lower index and four studies reported no significant differences. Due to the significant heterogeneity among studies (I2 = 97%, P < 0.05), we adopted the random effects model for analysis. The results showed no significant difference in the Chao1 index between patients with allergic rhinitis and healthy individuals (SMD = −0.00, 95% CI [−1.32–1.32]) (Fig. 2C).

The relative abundance of GM at phylum level

The main results of the relative abundance of gut microbiota at the phylum level are shown in Table S4.

Three studies have reported alterations in the relative abundance of Firmicutes in patients with allergic rhinitis (Zhu et al., 2020; Zhang et al., 2023; Wan et al., 2023). Significant heterogeneity was observed among the studies (I2 = 75%, P < 0.05), and a random-effects model was employed for the analysis. The relative abundance of Firmicutes was found to be similar between AR and healthy control (HC), with no significant difference observed (95% CI [−0.10–0.19]) (Fig. 3A).

Figure 3 Forest plot of relative abundance of gut microbiota at phylum level.

(A) Firmicutes, (B) Bacteroidetes, (C) Proteobacteria. (D) Actinobacteria.

Three studies reported alterations in the relative abundance of Bacteroidetes in patients with allergic rhinitis (Zhu et al., 2020; Liu et al., 2020; Wan et al., 2023). A significant degree of heterogeneity was observed among the studies (I2 = 95%, P < 0.05), and therefore a random effects modelling approach was utilised for the subsequent analysis. It was observed that the relative abundance of Bacteroidetes did not differ significantly between AR and HC (95% CI [−0.42–0.19]) (Fig. 3B).

The relative abundance of Proteobacteria was examined in three studies that reported changes in allergic rhinitis patients (Zhu et al., 2020; Liu et al., 2020; Wan et al., 2023). Significant heterogeneity was observed among studies (I2 = 92%, P < 0.05), thus a random-effects model was employed in the analysis. It was observed that the relative abundance of Proteobacteria did not exhibit a statistically significant difference between the AR and HC groups (95% CI [−0.06–0.03]) (Fig. 3C).

Alterations in the relative abundance of Actinobacteria among patients with allergic rhinitis have been reported in three studies (Zhu et al., 2020; Liu et al., 2020; Wan et al., 2023). Significant heterogeneity was observed across studies (I2 = 83%, P < 0.05), necessitating the use of a random effects model for analysis. Our findings indicate that the relative abundance of Actinobacteria does not exhibit a significant difference between AR and HC groups (95% CI [−0.09–0.03]) (Fig. 3D).

Subgroup analysis

The relationship between AR and HC categorized by age

To assess how age impacts the association between AR and healthy people, we conducted subgroup analysis on 10 studies. Studies were divided into age groups of children and adults. The results indicated that the subgroup analysis did not alter the previous findings. There was no significant difference in the alpha diversity (Shannon index, Chao1 index) of intestinal flora between allergic rhinitis patients and healthy people in adults and children (Fig. 4). The age factor was not the main source of heterogeneity in the study. The main results of the subgroup analysis categorized by age are shown in Table S5.

Figure 4 Subgroup analysis for age.

(A) Shannon. (B) Chao1.

The relationship between AR and HC categorized by region

To investigate how region impacts the association between AR and healthy people, we conducted subgroup analysis on 10 studies. Studies were divided into groups by different countries. The results indicated that the subgroup analysis did not alter the previous findings. There was no significant difference in the alpha diversity (Shannon index, Chao1 index) of intestinal flora between allergic rhinitis patients and healthy people from different regions (Fig. 5). The region factor was not the main source of heterogeneity in the study. The main results of the subgroup analysis categorized by region are shown in Table S6.

Figure 5 Subgroup analysis for region.

(A) Shannon. (B) Chao1.

The relationship between AR and HC categorized by measurement method

To investigate how measurement method impacts the association between AR and healthy people, we conducted subgroup analysis on 10 studies. Studies were divided into groups of 16S rRNA gene sequencing and shotgun metagenomic sequencing. The results indicated that the subgroup analysis did not alter the previous findings. There was no significant difference in the alpha diversity (Shannon index) of intestinal flora between allergic rhinitis patients and healthy people with different measurement method (Fig. 6). Due to a limited number of shotgun metagenomic studies, relevant subgroup analyses for Simpson index were not performed. However, heterogeneity within the shotgun metagenomic sequencing group decreased compared to the 16S rRNA group. The main results of the subgroup analysis categorized by measurement method are shown in Table S7.

Figure 6 Subgroup analysis for measurement method in Shannon index.

Sensitivity analysis

In order to ensure the reliability of the present meta-analysis, a sensitivity analysis was conducted to evaluate the robustness of the pooled results by eliminating each study at one time sequentially. This indicated that the heterogeneity among the studies was not significant in the Shannon index and Chao1 index (Fig. 7). Given the limited number of studies with other outcomes included, it was not possible to perform a sensitivity analysis.

Figure 7 The sensitivity analysis results of alpha diversity.

(A) Shannon. (B) Chao1.

Discussion

As far as we know, this is the inaugural meta-analysis to examine the gut microbiota of patients with allergic rhinitis and a control group. The gut microbiota is an integral part of the human body, influencing immunity, metabolism and function (Adak & Khan, 2019). Bacterial metabolites have also been linked to allergic rhinitis. Butyrate, mainly produced by Bacteroidetes and Firmicutes, has been observed to have a negative correlation with serum IgE levels and is thought to play a role in regulating the nasal mucosal epithelial barrier (Chiu et al., 2019b; Liu et al., 2023). Lipopolysaccharide, produced by gram-negative bacteria, has been demonstrated to induce the production of inflammatory factors, activate inflammatory cell pathways, and contribute to the development of allergic rhinitis (Mohr et al., 2022; Iwasaki et al., 2017). However, research findings indicate that high inter-individual variability in the gut microbiota may obscure subtle yet biologically significant associations. Additionally, differences in study design, including variations in sequencing techniques, sample processing methods, and data analysis approaches, may contribute to inconsistent results. Furthermore, unaccounted confounding factors, such as diet, medication use, and geographic location, could influence microbiota composition and obscure potential links between AR and gut microbiota (Hu et al., 2021). Therefore, a meta-analysis method was employed in order to systematically integrate and evaluate existing research evidence. This was undertaken with a view to comprehensively exploring the potential association between allergic rhinitis and intestinal flora. This systematic review and meta-analysis encompassed 10 studies comprising a total of 935 individuals. Five demonstrated no observed alpha diversity difference, two exhibited an increase, and three showed a decrease. Nevertheless, the results of the meta-analysis indicated that these changes were not statistically significant. The direction of the observed effects was inconsistent across studies, and the effect sizes were relatively small. Changes in overall microbial α diversity are a common indicator of ecological dysregulation (Avalos-Fernandez et al., 2022; Flint et al., 2012). Previous studies have also corroborated the absence of a difference in α diversity in certain other diseases. This indicates that the detection methods and cognition of gut flora are still limited, and that the gut flora varies greatly among individuals. For instance, Cao, Gao & Huang (2024) observed no significant differences in α diversity between patients with osteoporosis and healthy individuals. Furthermore, authors suggest that the results may be limited by the high heterogeneity of the study. In both articles describing the increased Shannon index, an increase in the bacterial community of Firmicutes in patients with allergic rhinitis was observed, as well as an increase in the relative abundance of Firmicutes (Zhu et al., 2020; Wan et al., 2023). However, two additional studies examining the relative abundance of Firmicutes reached opposing conclusions, and the diversity conclusions of the two studies also contradicted each other (Watts et al., 2021; Zhou et al., 2021). These inconsistencies may be attributed to differences in study populations, as microbiota composition is influenced by factors such as age, diet, and geographic location. Additionally, methodological discrepancies, such as sequencing platforms, bioinformatics pipelines, and statistical approaches, may have contributed to the divergent findings. Future research should employ standardized methodologies and consider potential confounding variables to enhance comparability across studies. Further research is required to ascertain whether the abundance of Firmicutes directly influences the diversity outcomes.

A total of four main phylum were extracted from the data set, namely Firmicutes, Bacteroidetes, Proteobacteria and Actinobacteria (Qin et al., 2010). In the meta-analysis, no statistically significant differences were found in the abundance of various phyla between patients with AR and the control group. While the studies indicated alterations in bacterial abundance, these findings were inconsistent, raising questions about the representativeness of microbiota results due to variations across populations and regions. The core microbiota tends to be stable; however, the diversity and abundance of bacterial species in the human gut can differ significantly among individuals and geographic areas. These individual differences in microbiota composition may arise from a variety of factors, including genetics, dietary patterns, and environmental influences (Das & Nair, 2019; Nicholson et al., 2012). A longitudinal one-year population study found that a total of 23% of the compositional variance was explained by intra-individual variation (Olsson et al., 2022). If the included studies did not adequately control for these variables or if they were not entirely excluded, it is plausible that potential differences in microbiota related to AR were obscured. The significant variability in gut microbiota among individuals could mask subtle yet relevant associations between AR and microbiota composition. To enhance the reliability of findings regarding the microbiota and AR, future studies should adopt more rigorous designs that account for individual and population differences. Investigating specific dietary patterns, lifestyle factors, and environmental exposures could yield insights into how these variables influence the microbiota’s role in AR pathogenesis. Additionally, longitudinal studies that track microbiota changes over time in relation to AR development may provide a clearer understanding of causality.

In order to facilitate the objectives of this study, it is necessary to group participants according to age and region. There are discrepancies in immune system development and intestinal flora composition among different age groups. For example, children have developing immune systems and dynamic intestinal flora, which may lead to different allergic rhinitis pathogenesis and flora interactions compared to adults (Derrien, Alvarez & De Vos, 2019). Age-based subgroup analysis helps clarify these relationships, aiding in the development of targeted prevention and treatment strategies. The composition and diversity of intestinal flora are influenced by various environmental, dietary, and climatic factors, which in turn can affect the occurrence and development of allergic rhinitis. For instance, the intestinal flora structure of populations may differ in humid and dry climates, and the incidence and severity of allergic rhinitis may also differ. Through regional subgroup analysis, the influence of different regional factors on the relationship between the two can be clarified, which is helpful to formulate the prevention and control strategy according to local conditions. Most studies focus on China, where geographical and climatic diversity complicates regional analysis (Huang et al., 2024). To enhance the robustness of the findings and ensure the relevance to other regions, further studies with more extensive sample sizes are necessary.

The research method also has a major impact on the accuracy of gut microbiota results. Differences in fecal sample collection (e.g., home-based vs. clinic-based collection), storage conditions (e.g., immediate freezing vs. room temperature storage), and sequencing technologies (e.g., 16S rRNA vs. shotgun metagenomics) can introduce variability in results. Furthermore, factors such as sequencing depth and bioinformatics processing pipelines may influence taxonomic resolution and abundance estimates. Standardization of microbiota research methodologies, including sample handling and data processing, is crucial to improve the reliability and reproducibility of findings. The development of more advanced technical means is the main approach of researchers. 16S rRNA sequencing technology has long been regarded as a pivotal tool for analysing the structure and diversity of gut microbial communities (Shahi et al., 2019). Nevertheless, with the rapid advancement of biotechnology and continuous innovation in scientific research, metagenomic sequencing, a more comprehensive and in-depth method, has gradually gained attention from researchers and is increasingly being applied in various types of research (Shi et al., 2022). The three studies that employed metagenomic sequencing were all published in 2023 (Lin, Li & Li, 2022; Chiu et al., 2023; Wan et al., 2023). Although metagenomic sequencing is primarily focused on omics, it still enables the extraction and analysis of essential data. A comparison of subgroup analysis data from 16S rRNA sequencing and metagenomic sequencing revealed that both subgroups obtained similar results in analysing microbial community structure. Nevertheless, metagenomic sequencing exhibited superior heterogeneity with a lower degree of variation. This finding may be attributed to the continuous development and optimisation of metagenomic sequencing technology. As sequencing technology continues to evolve, the accuracy and specificity of metagenomic sequencing have improved significantly, enabling more precise resolution of microbial community complexity and diversity (Ranjan et al., 2016).

This study did not include sufficient data to directly analyze the changes in β diversity and microbial metabolic functions of the gut microbiota in patients with allergic rhinitis. However, existing literature suggests that both may play important roles in the pathogenesis of the disease. β diversity reflects differences in microbial community structure, and the gut microbiota of allergic rhinitis patients may exhibit significantly different compositional patterns compared to healthy individuals. These alterations extend beyond the overall structure of the microbiota, encompassing shifts in the abundance of specific genera or species. Moreover, these shifts in β-diversity may be associated with the immune status and inflammatory levels of the host, thereby impacting the progression and severity of allergic rhinitis (Liu et al., 2024). Concurrently, these disparities may further impact microbial metabolic functions, including the production of short-chain fatty acids (SCFAs). SCFAs, as pivotal microbial metabolites, have been demonstrated to play a pivotal role in regulating immune responses and maintaining intestinal barrier function (Wang et al., 2023). Consequently, subsequent studies should persist in exploring the alterations of intestinal microbiota β-diversity and microbial metabolic function in patients with allergic rhinitis as research directions to procure additional evidence to support further analysis. This comprehensive approach will facilitate a more profound comprehension of the mechanisms through which gut microbiota contributes to the development of allergic rhinitis.

The International Consensus Statement on Allergy and Rhinology (ICSAR) recommends the consideration of probiotics as an adjuvant therapy for patients with AR due to their minimal harm and proven efficacy in improving symptoms (Wise et al., 2023). However, other studies have made different points. The meta-analysis of the efficacy of probiotics on AR demonstrated that probiotics are effective in improving the symptoms and quality of life of patients with allergic rhinitis (Yan et al., 2022). Nevertheless, no significant differences were observed in blood eosinophil counts and antigen-specific serum IgE levels between the placebo and probiotic groups. A review published in 2009 that evaluated the efficacy of different treatments approved in the United States for allergic rhinitis documented a high placebo response strength (Benninger et al., 2010). The overall symptom score improved by 15% in seasonal allergic rhinitis and 24.8% in perennial allergic rhinitis. A recently published randomised controlled trial demonstrated that open-label placebos were more efficacious than conventional treatments in improving allergy symptoms (Schaefer, Harke & Denke, 2016; Schaefer, Zimmermann & Enck, 2023). This may be attributed to the subjective influence of patients on the outcome variables in judging the symptoms of allergic rhinitis. These discrepancies may arise from differences in probiotic strains, dosages, treatment durations, and patient populations. Moreover, host factors, such as baseline microbiota composition and immune status, may influence probiotic efficacy (Fentie et al., 2024). Further well-designed randomized controlled trials with standardized probiotic formulations are needed to determine their role in AR management.

This meta-analysis offers valuable insights into the relationship between gut microbiota and AR, marking a significant contribution to the existing literature by systematically reviewing data from multiple studies. One of its strengths is the comprehensive analysis of diverse research findings, allowing for a broader understanding of microbial diversity and its potential implications for AR. However, several limitations should be noted. Firstly, it should be noted that only six databases were searched in this study. Consequently, the search may be incomplete. Future research should expand the number of databases searched and include grey literature to ensure a more comprehensive literature review and reduce the risk of publication bias. Secondly, the difficulty in obtaining raw data for all included studies precluded the performance of a consolidated analysis for additional metrics, such as beta diversity metrics and the relative abundance of GM at genus level. To overcome this, future research should focus on collaborating with original study authors for data access and establishing standardized protocols for data sharing, which could enhance the comprehensiveness of future analyses. Thirdly, some studies require the manual extraction of certain basic data from the graph, which may introduce a further form of bias. Nevertheless, the process of data extraction was subjected to extensive discussion and a consensus was reached by two reviewers. It is therefore assumed that any potential impact on inter-group comparisons will remain insignificant, given that this approach was consistently applied across all studies. The present study only discusses the structure and composition of GM, without in-depth analysis of the transcriptomics and proteomics of GM function. Fourth, the total sample size involved in the study is relatively small, and the sample size of the included studies is also small, which may not be representative of all patients. It is possible that our meta-analysis is underpowered, and that further studies are required to validate our findings in a larger population.

Conclusion

The present study did not identify any notable discrepancies in the alpha diversity and phyla-level bacterial abundance of the intestinal flora between AR patients and healthy individuals. The necessity of probiotics in modifying the composition of the gut microbiota remains to be elucidated. Additional studies are needed to deepen the understanding of the relationship between gut microbiota dysregulation and allergic rhinitis.

Supplemental Information

Supplemental Information 1 Stool sample processing methods in the included studies

Supplemental Information 2 Search terms for each electronic database

Supplemental Information 3 Main results of alpha diversity

Supplemental Information 4 Main results of relative gut microbiota abundance at phylum level

Supplemental Information 5 Main results of subgroup analysis categorized by age

Supplemental Information 6 Main results of subgroup analysis categorized by region

Supplemental Information 7 Main results of subgroup analysis categorized by measurement method

Supplemental Information 8 PRISMA checklist

Supplemental Information 9 Intended audience

Additional Information and Declarations

Competing Interests

Author Contributions

Data Availability

The authors declare there are no competing interests.

Mengyao Li conceived and designed the experiments, performed the experiments, analyzed the data, prepared figures and/or tables, authored or reviewed drafts of the article, and approved the final draft.

Qian Wang conceived and designed the experiments, performed the experiments, analyzed the data, prepared figures and/or tables, authored or reviewed drafts of the article, and approved the final draft.

Ruikun Wang conceived and designed the experiments, prepared figures and/or tables, authored or reviewed drafts of the article, and approved the final draft.

Jian Pu conceived and designed the experiments, prepared figures and/or tables, authored or reviewed drafts of the article, and approved the final draft.

Yimin Zhang performed the experiments, authored or reviewed drafts of the article, and approved the final draft.

Siyu Ye performed the experiments, authored or reviewed drafts of the article, and approved the final draft.

Jieqiong Liang analyzed the data, prepared figures and/or tables, and approved the final draft.

Tao Li conceived and designed the experiments, authored or reviewed drafts of the article, and approved the final draft.

Qinglong Gu conceived and designed the experiments, authored or reviewed drafts of the article, and approved the final draft.

The following information was supplied regarding data availability:

This is a systematic review/meta-analysis.

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
