# Peer review of "Association between gut microbiota and allergic rhinitis: a systematic review and meta-analysis"

_PeerJ, doi:10.7717/peerj.19441_

## Round 0.1 · original submission · Minor Revisions

The submitted manuscript is well written and was positively evaluated by the reviewers. Based on the reviewers assessment, I recommend the manuscript for Minor revision.

·

Basic reporting

Overall, the paper is self-contained and well-organized. The academic writing is professional and efficient. The figures and tables are professionally designed and effectively convey the results.

Some points for improvement:

1.There are some typos that need correction (e.g., line 93).
2.The authors should clarify that "HC" stands for "healthy control" in the text. It would be good to clarify "AR" and "HC" in the tables and figures as well.
3.The figures in the PDF file are not clear enough.

Experimental design

The overall design of the study is good and clearly explained in the paper. Nearly every step of the meta-analysis is conducted thoroughly, with additional people checking for errors. The use of statistical tools seems legitimate to me. I appreciate the subgroup analysis and the use of a random-effects model to account for heterogeneity among the studies.

Validity of the findings

The conclusions based on the analysis seem legitimate and of high quality to me. The studies from which the authors extract data are listed in the references and tables, which appears well-organized to me.

Reviewer 2 ·

Basic reporting

In recent years, plenty of information has been compiled about microbiota’s significant impact on human health. Its ability to modulate immune system response remains one of the most popular issues of 21st century. Studies suggest that the perturbations in the intestinal microbes’ composition may increase the risk of allergy, however, not all studies agree on this point. The authors of the reviewed article attempted to summarize the research conducted. The authors prepared the first meta-analysis comparing the gut microbiota of patients with allergic rhinitis and a control group.The meta-analysis was conducted appropriately, I have no specific comments.

Experimental design

No comment.

Validity of the findings

It seems that the authors assessed all relevant research in this area. Conclusions are well stated, linked to original research question. Conclusions are important and suggest directions for further research.

Reviewer 3 ·

Basic reporting

Review Comments:
The manuscript presents a comprehensive systematic review and meta-analysis on an important topic. The inclusion of recent studies up to December 2023 ensures the review's relevance and currency. The authors have conducted a thorough quality assessment of the included studies and employed appropriate statistical analysis methods, enhancing the robustness of their findings. However, several areas for improvement are identified. By addressing these points, the manuscript could make a stronger contribution to the field and provide more valuable insights for readers.

1. The introduction needs more detail and context. I suggest expanding the background information on the gut microbiota and its potential role in allergic rhinitis. Specifically:
- Provide more details on the current understanding of gut microbiota dysbiosis and its systemic effects
- Include recent findings on the gut-nose axis and its implications for allergic diseases
- Strengthen the rationale for conducting this meta-analysis.

2. **Results Presentation: Enhance Data Visualization**
- Improve the clarity of forest plots (Figures 2-7) by:
* Increasing font sizes for better readability
* Adding clearer labels for subgroups
* Using consistent color schemes across figures
* Including effect size indicators on the plots
- Consider adding a table summarizing the main findings of the meta-analysis

3. **Language and Formatting: Improve Readability**
- The manuscript would benefit from professional English editing to:
* Improve sentence structure and flow
* Ensure consistent use of terminology
* Correct grammatical errors
* Enhance overall readability for an international audience
- Standardize formatting throughout the manuscript, particularly in:
* Reference formatting
* Figure and table captions
* Statistical reporting
-Line 93: The manuscript contains a notable spelling error at the beginning of the sentence: "Tis" should be corrected to "This."
-There are a few typographical errors throughout the text, such as "A total often studies" instead of "A total of ten studies" (line 178).

Experimental design

- Clarify the inclusion/exclusion criteria for studies, particularly regarding age ranges and diagnostic criteria for AR. Although this paper presents detailed criteria for inclusion in the ranking, the rationale for the final deletion of more than 5,000 non-duplicate articles is not clear (lines 155-157), and it would be more convincing to provide a more detailed explanatory statement.
-Enhance Subgroup Analysis Presentation: The subgroup analyses based on age, region, and measurement method are commendable. However, the presentation of these results could be improved by including more detailed tables or figures that clearly show the subgroup classifications, sample sizes, and specific statistical outcomes.
-The article mentions several times that “the results of human observational studies on the correlation between intestinal flora and AR have been contradictory”. but the references you cite do not support your point, e.g., ref. 26, etc.

Validity of the findings

*Discussion: Strengthen Interpretation and Contextualization**
- Expand the discussion to better interpret the null findings
- Discuss potential reasons for the high heterogeneity observed
- Discuss Inconsistencies in Study Findings: The results section notes inconsistencies in the direction of differences across studies for various indices. A detailed discussion on these inconsistencies and potential reasons for them would be beneficial. This could involve considering differences in study design, sample size, or the specific microbiome features being assessed.
--The discussion in this paper only discusses the inclusion of the database's and the amount of data on the conclusions, and does not discuss the deeper possible reasons for the conclusions of this experiment, such as the need for age strata, etc., and requires the authors to provide more possible explanations or discussions to support your conclusions.

-Consider Additional Indices and Outcomes: The manuscript focuses on alpha diversity indices such as Shannon and Chao1. While these indices are commonly used, considering additional indices (e.g., Simpson index, beta diversity measures) or outcomes (e.g., specific microbial taxa, metabolic pathways) could provide a more comprehensive view of the intestinal flora changes in AR patients. If relevant studies using these additional indices or outcomes exist, the authors could consider including them in their meta-analysis or at least discuss their potential relevance.

Additional comments

- Ensure all supplementary tables and figures are clearly labeled and referenced in the main text
- Provide more detailed descriptions of the supplementary materials

·

Basic reporting

Dear Authors, thanks for submitting such an inaugural meta-analysis articles review to examine the gut microbiota of patients with allergic rhinitis and a control group. Well written article, not need any major or minor reviews. Well done.
Warm regards

Experimental design

Practical experimental design

Validity of the findings

Concede with the disease behavior

---

## Round 0.2 · accepted · Accept

The manuscript improved a lot after the first revision. After a second revision to ensure the best possible quality of the manuscript, the reviewers recommended "accept". On behalf of the reviewing process, I support this recommendation; the revised manuscript is accepted for publication.

·

Basic reporting

I appreciate the authors’ efforts in addressing my earlier feedback. I have no additional comments at this time.

Experimental design

I appreciate the authors’ efforts in addressing my earlier feedback. I have no additional comments at this time.

Validity of the findings

I appreciate the authors’ efforts in addressing my earlier feedback. I have no additional comments at this time.